# WAFFLe: Weight Anonymized Factorization for Federated Learning

## Abstract

In domains where data are sensitive or private, there is great value in methods that can learn in a distributed manner without the data ever leaving the local devices. In light of this need, federated learning has emerged as a popular training paradigm. However, many federated learning approaches trade transmitting data for communicating updated weight parameters for each local device. Therefore, a successful breach that would have otherwise directly compromised the data instead grants whitebox access to the local model, which opens the door to a number of attacks, including exposing the very data federated learning seeks to protect. Additionally, in distributed scenarios, individual client devices commonly exhibit high statistical heterogeneity. Many common federated approaches learn a single global model; while this may do well on average, performance degrades when the i.i.d. assumption is violated, underfitting individuals further from the mean and raising questions of fairness. To address these issues, we propose Weight Anonymized Factorization for Federated Learning (WAFFLe), an approach that combines the Indian Buffet Process with a shared dictionary of weight factors for neural networks. Experiments on MNIST, FashionMNIST, and CIFAR-10 demonstrate WAFFLe's significant improvement to local test performance and fairness while simultaneously providing an extra layer of security.

## 1 Introduction

With the rise of the Internet of Things (IoT), the proliferation of smart phones, and the digitization of records, modern systems generate increasingly large quantities of data. These data provide rich information about each individual, opening the door to highly personalized intelligent applications, but this knowledge can also be sensitive: images of faces, typing histories, medical records, and survey responses are all examples of data that should be kept private. Federated learning (McMahan et al., 2017) has been proposed as a possible solution to this problem. By keeping user data on each local *client* device and only sharing model updates with the global *server*, federated learning represents a possible strategy for training machine learning models on heterogeneous, distributed networks in a privacy-preserving manner. While demonstrating promise in such a paradigm, a number of challenges remain for federated learning (Li et al., 2019).

As with centralized distributed learning settings (Dean et al., 2012), many federated learning algorithms focus on learning a single global model. However, due to variation in user characteristics or tendencies, personal data are highly likely to exhibit significant *statistical heterogeneity*. To simulate this, federated learning algorithms are commonly tested in non-i.i.d. settings (McMahan et al., 2017; Smith et al., 2017; Li & Wang, 2019; Peterson et al., 2019), but data are often equally represented across clients and ultimately a single global model is typically learned. As is usually the case for one-size-fits-all solutions, while the model may perform acceptably on average for many users, some clients may see very poor performance. Questions of fairness (Mohri et al., 2019; Li et al., 2020) may arise if performance is compromised for individuals in the minority in favor of the majority.

Another challenge for federated learning is security. Data privacy is the primary motivation for keeping user data local on each device, rather than gathering it in a centralized location for training. In traditional distributed learning systems, data are exposed to additional vulnerabilities while being transmitted to and while residing in the central data repository. In lieu of the data, many federated learning approaches require clients to send weight updates to train the aggregated model.

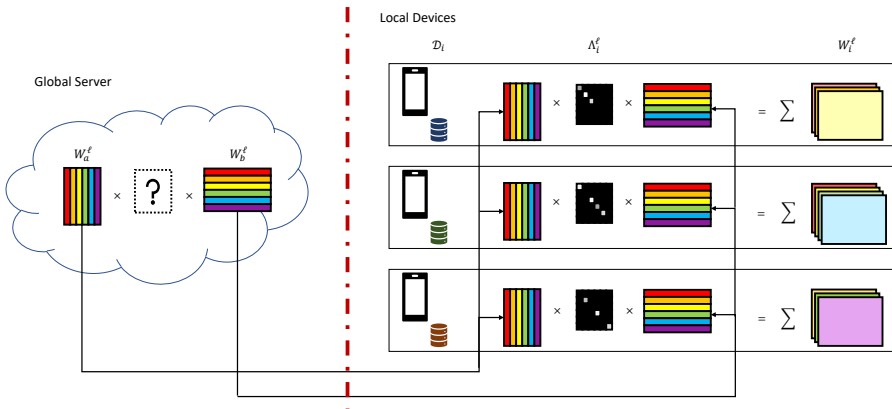

Figure 1: In WAFFLe, the clients share a global dictionary of rank-1 weight factors $\{W_a^\ell, W_b^\ell\}$. Each client uses a sparse diagonal matrix $\Lambda_i^\ell$, specifying the combination of weight factors that constitute its own personalized model. Neither the client data $\mathcal{D}_i$ nor factor selections $\Lambda_i^\ell$ leave the local device.

However, the threat of membership inference attacks (Shokri et al., 2017; Nasr et al., 2019) or model inversion (Fredrikson et al., 2015; Zhu et al., 2019) mean that private data on each device can still be compromised if federated learning updates are intercepted or if the central server is breached.

We propose **W**eight **A**nonymized **F**actorization for **F**ederated **Le**arning (WAFFLe), leveraging Bayesian nonparametrics and neural network weight factorization to address these issues. Rather than learning a single global model, we learn a dictionary of rank-1 weight factor matrices. By selecting and weighting these factors, each local device can have a model customized to its unique data distribution, while sharing the learning burden of the weight factors across devices. We employ the Indian Buffet Process (Ghahramani & Griffiths, 2006) as a prior to encourage factor sparsity and reuse of factors, performing variational inference to infer the distribution of factors for each client. While updates to the dictionary of factors are transmitted to the server, the distribution capturing which factors a client uses are kept local. This adds an extra insulating layer of security by obfuscating which factors a client is using, hindering an adversary's ability to perform membership inference attacks or dataset reconstruction.

We perform experiments on MNIST (LeCun et al., 1998), FMNIST (Xiao et al., 2017), and CIFAR-10 (Krizhevsky, 2009) in settings exhibiting strong statistical heterogeneity. We observe that the model customization central to WAFFLe's design leads to higher performance for each client's local distribution, while also being significantly fairer across all clients. Finally, we perform membership inference (Shokri et al., 2017) and model inversion (Fredrikson et al., 2015) attacks on WAFFLe, showing that it is much harder to expose user data than with FedAvg (McMahan et al., 2017).

## 2 METHODOLOGY

### 2.1 LEARNING A SHARED DICTIONARY OF WEIGHT FACTORS

**Single Global Model** Consider $N$ client devices, with the $i^{\text{th}}$ device having data distribution $\mathcal{D}_i$, which may differ as a function of $i$. In many distributed learning settings, a single global model is learned and deployed to all $N$ clients. Thus, assuming a multilayer perceptron (MLP) architecture[1] with layers $\ell = 1, ..., L$, the set of weights $\theta = \{W^\ell\}_{\ell=1}^L$ is shared across all clients. To satisfy the global objective, $\theta$ is learned to minimize the loss on average across all clients. This is the approach of many federated learning approaches. For example, FedAvg (McMahan et al., 2017) minimizes the following objective:

$$\min_\theta \mathscr{L}(\theta) = \sum_{i=1}^N p_i \mathcal{L}_i(\theta) \qquad (1)$$

where $\mathcal{L}_i(\theta) \coloneqq \mathbb{E}_{x_i \sim \mathcal{D}_i}[l_i(x_i; \theta)]$ is the local objective function, $N$ is the number of clients, and $p_i \geq 0$ is the weight of each device $i$. However, given statistical heterogeneity, such a one-size-fits-all

---

[1]While we restrict our discussion to fully connected layers here for simplicity, this can be generalized to other types of layers as well. See Appendix A for 2D *convolutional* layers.

approach may lead to the global model underfitting on certain clients; often this translates to how close a particular client's local distribution is to the population distribution. As a result, this model may be viewed as less fair to these clients with less common traits.

**Individual Local Models** On the other extreme, we may alternatively consider learning $N$ local models $\theta_i = \{W_i^\ell\}_{\ell=1}^L$, each only trained on $\mathcal{D}_i$. In this case, each set of weights $\theta_i$ is maximally specific to the data distribution of each client $i$. However, each client typically has limited data, which may be insufficient for training a full model without overfitting; the total number of parameters that must be learned across all clients scales with $N$. Additionally, learning $N$ separate models does not take advantage of similarities between client data distributions or the shared learning task.

**Shared Weight Factors** To make more efficient use of data, we instead propose a compromise between a single global model and $N$ individual local models. Specifically, we allow each client's model to be personalized to the client's local distribution, but with all models sharing a dictionary of jointly learned components. Using a layer-wise decomposition (Mehta et al., 2020), we construct each weight matrix with the following factorization:

$$W_i^\ell = W_a^\ell \Lambda_i^\ell W_b^\ell \tag{2}$$

$$\Lambda_i^\ell = \mathrm{diag}(\boldsymbol{\lambda}_i^\ell) \tag{3}$$

where $W_a^\ell \in \mathbb{R}^{J \times F}$ and $W_b^\ell \in \mathbb{R}^{F \times M}$ are global parameters shared across clients and $\boldsymbol{\lambda}_i^\ell \in \mathbb{R}^F$ is a client-specific vector. This factorization can be equivalently expressed as

$$W_i^\ell = \sum_{k=1}^F \lambda_{i,k}^\ell \left( \boldsymbol{w}_{a,k}^\ell \otimes \boldsymbol{w}_{b,k}^\ell \right) \tag{4}$$

where $\boldsymbol{w}_{a,k}^\ell$ is the $k^{\text{th}}$ column of $W_a^\ell$, $\boldsymbol{w}_{b,k}^\ell$ is the $k^{\text{th}}$ row of $W_b^\ell$, and $\otimes$ represents an outer product. Written in this way, the interpretation of the corresponding pairs of columns and rows $\boldsymbol{w}_{a,k}^\ell$ and $\boldsymbol{w}_{b,k}^\ell$ as weight *factors* is more apparent: $W_a^\ell$ and $W_b^\ell$ together comprise a global dictionary of the weight factors, and $\boldsymbol{\lambda}_i^\ell$ can be viewed as the factor *scores* of client $i$. Differences in $\boldsymbol{\lambda}_i^\ell$ between clients allows for customization of the model to each client's data distribution (see Figure 1), while sharing of the underlying factors $W_a^\ell$ and $W_b^\ell$ enables learning from the data of all clients.

We constitute each of the client's factor scores $\boldsymbol{\lambda}_i^\ell$ as the element-wise product:

$$\boldsymbol{\lambda}_i^\ell = \boldsymbol{r}^\ell \odot \boldsymbol{b}_i^\ell \tag{5}$$

where $\boldsymbol{r}^\ell \in \mathbb{R}^F$ indicates the strength of each factor and $\boldsymbol{b}_i^\ell \in \{0,1\}^F$ is a binary vector indicating the active factors. As explained below, $\boldsymbol{b}_i^\ell$ is typically sparse, so in general each client only uses a small subset of the available weight factors. Throughout this work, we use the absence of the $\ell$ superscript (*e.g.*, $\boldsymbol{\lambda}_i$) to refer to the entire collection across all layers for which this factorization is done. We learn a point-estimate for $W_a$, $W_b$ and $\boldsymbol{r}$.

## 2.2 The Indian Buffet Process

**Desiderata** Within the context of federated learning with statistical heterogeneity, there are a number of desirable properties we wish the client factor scores to have collectively. Firstly, $\boldsymbol{\lambda}_i$ should be *sparse*, which encourages consolidation of related knowledge while minimizing interference: client A should be able to update the global factors during training without destroying client B's ability to perform its own task. This encourages *fairness*, as in settings with multiple subpopulations, this interference is most likely to be at the smaller groups' expense. On the other hand, we would also like factors to be reused among clients. While data may be non-i.i.d. across clients, there are often some similarities or overlap; thus, *shared* factors distribute learning across all clients' data, avoiding the $N$ independent model's scenario. Finally, in the distributed settings considered in federated learning, the total number of nodes is rarely pre-defined. Therefore, there needs to be a way to gracefully *expand* to accommodate new clients to the system without re-initializing the whole model. This includes both increasing server-side capacity if necessary and initializing new clients.

**Prior** Given these desiderata, the Indian Buffet Process (IBP) (Ghahramani & Griffiths, 2006) is a natural choice. As a prior, the IBP regularizes client factors to be sparse, and new factors are introduced but at a harmonic rate, preferring reusing factors as much as possible over initializing new ones. This Bayesian nonparametric approach allows the data to dictate client factor assignment,

---

**Algorithm 1** Weight Anonymized Factorization for Federated Learning (WAFFLe).

---
1: **Input:** Communication rounds $T$, local training epochs $E$, learning rate $\eta$
2: Server initializes global weight factor dictionaries $W_a$ and $W_b$, factor strengths $\boldsymbol{r}$
3: Clients each initialize variational parameters $\boldsymbol{\pi}_i, \boldsymbol{c}_i, \boldsymbol{d}_i$
4: **for** $t = 1, \cdots, T$ **do**
5:     Server randomly selects subset $\mathcal{S}_t$ of clients and sends $\{W_a, \boldsymbol{r}, W_b\}$
6:     **for** client $i \in \mathcal{S}_t$ **in parallel do**
7:         $W_a, \boldsymbol{r}, W_b, \boldsymbol{\pi}_i, \boldsymbol{c}_i, \boldsymbol{d}_i \leftarrow$ CLIENTUPDATE$(W_a, \boldsymbol{r}, W_b, \boldsymbol{\pi}_i, \boldsymbol{c}_i, \boldsymbol{d}_i)$
8:         Send $\{W_a, \boldsymbol{r}, W_b\}$ to the server.
9:     **end for**
10:    Server aggregates and averages updates $\{W_a, \boldsymbol{r}, W_b\}$
11: **end for**

12: **function** CLIENTUPDATE$(W_a, \boldsymbol{r}, W_b, \boldsymbol{\pi}_i, \boldsymbol{c}_i, \boldsymbol{d}_i)$
13:     **for** $e = 1, \cdots, E$ **do**
14:         **for** minibatch $b \in \mathcal{D}_i$ **do**
15:             Update $\{W_a, \boldsymbol{r}, W_b, \boldsymbol{\pi}_i, \boldsymbol{c}_i, \boldsymbol{d}_i\}$ by minimizing (12)
16:         **end for**
17:     **end for**
18:     Return $\{W_a, \boldsymbol{r}, W_b, \boldsymbol{\pi}_i, \boldsymbol{c}_i, \boldsymbol{d}_i\}$
19: **end function**

---

factor reuse, and server-side model expansion. We use the stick-breaking construction of the IBP as a prior for factor selection:

$$v_{i,\kappa}^\ell \sim \text{Beta}(\alpha, 1) \tag{6}$$

$$\pi_{i,k}^\ell = \prod_{\kappa=1}^{k} v_{i,\kappa}^\ell \tag{7}$$

$$b_{i,k}^\ell \sim \text{Bernoulli}(\pi_{i,k}^\ell) \tag{8}$$

with $\alpha$ a hyperparameter controlling the expected number of active factors and the rate of new factors being incorporated, and $k$ indexes the factor.

**Inference** We learn the posterior distribution for the random variables $\phi_i = \{\boldsymbol{b}_i, \boldsymbol{v}_i\}$. Exact inference of the posterior is intractable, so we employ variational inference with mean-field approximation to determine the active factors for each client device, using the following variational distributions:

$$q(\boldsymbol{b}_i^\ell, \boldsymbol{v}_i^\ell) = q(\boldsymbol{b}_i^\ell)q(\boldsymbol{v}_i^\ell) \tag{9}$$

$$\boldsymbol{b}_i^\ell \sim \text{Bernoulli}(\boldsymbol{\pi}_i^\ell) \tag{10}$$

$$\boldsymbol{v}_i^\ell \sim \text{Kumaraswamy}(\boldsymbol{c}_i^\ell, \boldsymbol{d}_i^\ell) \tag{11}$$

learning the variational parameters $\{\boldsymbol{\pi}_i, \boldsymbol{c}_i, \boldsymbol{d}_i\}$ for each queried client using Bayes by Backprop (Blundell et al., 2015). Needing a differentiable parameterization, we use the Kumaraswamy distribution (Kumaraswamy, 1980) as a replacement for the Beta distribution of $\boldsymbol{v}_i$ and utilize a soft relaxation of the Bernouilli distribution (Maddison et al., 2017). The objective for each client is to maximize the variational lower bound:

$$\mathcal{L}_i(\theta) = \sum_{n=1}^{|\mathcal{D}_i|} \mathbb{E}_q \log p\left(y_i^{(n)} \big| \phi_i, x_i^{(n)}, W_a, W_b, \boldsymbol{r}\right) - \underbrace{\text{KL}\left(q\left(\phi_i\right) || p\left(\phi_i\right)\right)}_{\mathscr{R}} \tag{12}$$

$$\mathscr{R} = \sum_{\ell=1}^{L} \text{KL}\left(q(\boldsymbol{b}_i^\ell)||p(\boldsymbol{b}_i^\ell|\boldsymbol{v}_i^\ell)\right) + \text{KL}\left(q(\boldsymbol{v}_i^\ell)||p(\boldsymbol{v}_i^\ell)\right) \tag{13}$$

where $\theta = \{W_a, W_b, \boldsymbol{r}, \boldsymbol{b}_i\}$ and $|\mathcal{D}_i|$ is the number of training examples at client $i$. Note that in (12) the first term provides label supervision and the second term ($\mathscr{R}$) regularizes the posterior not to stray far from the IBP prior.

### 2.3 CLIENT-SERVER COMMUNICATION

**Training** Before the training begins, the global weight factors $\{W_a, W_b\}$ and the factor strengths $r$ are initialized by the server. Once initialized, each training round begins with $\{W_a, W_b, r\}$ being sent to the selected subset of clients. Each sampled client then trains the model on their own private dataset $\mathcal{D}_i$ for $E$ epochs, updating not only the weight factor dictionary $\{W_a, W_b\}$ and the factor strengths $r$, but also its also own variational parameters $\{\pi_i, c_i, d_i\}$, which controls which factors it uses. Once local training is finished, each client sends $\{W_a, W_b, r\}$ back to the server, but not $\{\pi_i, c_i, d_i\}$, which remain with the client with data $\mathcal{D}_i$. After the server has received back updates from all clients, the various new values for $\{W_a, W_b, r\}$ are aggregated with a simple averaging step. The process then repeats, with the server selecting a new subset of clients to query, sending the new updated set of global parameters, until the desired number of communication rounds have passed. This process is summarized in Algorithm 1.

**Evaluation** When a client enters the evaluation mode, it requests the current version of global parameters $\{W_a, W_b, r\}$ from the server. If the client has been previously queried for federated training, the local model consists of the aggregated global parameters and the factor score vector generated by its own local variational parameters $\{\pi_i\}$. Otherwise, the client uses only the aggregated $\{W_a, W_b, r\}$. Note that if a client has been previously queried, the most recently cached copy of the global parameters is an option if a network connection is unavailable or too expensive; in our experiments, we assume clients are able to request the most up-to-date parameters.

**Security** Data security is one of the central tenets of federated learning. Simpler, more standard methods of training a model could be utilized if all data were first aggregated at a central server. However, sensitive client data being intercepted during transmission or the server's data repository being breached by an attacker are major concerns, motivating federated learning's approach of keeping the data on the local device. On the other hand, keeping the data client-side may not be sufficient. Just as data can be compromised in transit or at the central database in non-federated settings, federated training updates are similarly vulnerable. In methods like FedAvg, this update is the entirety of the model's parameters. Effectively, this means that FedAvg trades yielding the data immediately for surrendering whitebox access to the model, which opens the model to a wide range of malicious activities (Szegedy et al., 2014; Fredrikson et al., 2015; Shokri et al., 2017; Wang et al., 2019; Zhu et al., 2019), including, critically, exposing the very data that federated learning aims to protect. With WAFFLe, clients transmit back the entire dictionary of weight factors $\{W_a, W_b\}$ and $r$, but not $\{\pi_i, c_i, d_i\}$. As such, the knowledge of which specific factors that a particular client uses is kept local. Therefore, even if messages are intercepted, an adversary cannot completely reconstruct the model, hampering their ability to perform attacks to recover the data.

## 3 RELATED WORK

### 3.1 STATISTICAL HETEROGENEITY

Statistical heterogeneity of the data distributions of client devices has long been recognized as a challenge for federated learning. Despite acknowledging statistical heterogeneity, many federated learning algorithms focus on learning a single global model (McMahan et al., 2017); such an approach often suffers from model divergence, as local models may vary significantly from each other. To address this, a number of works break away from the single-global-model formulation. Several (Smith et al., 2017; Corinzia & Buhmann, 2019) have cast federated learning as a multi-task learning problem, with each client treated as a separate task. FedProx (Li et al., 2018) adds a proximal term to account for statistical heterogeneity by limiting the impact of local updates. Others study federated learning within a model-agnostic meta-learning framework (Jiang et al., 2019; Khodak et al., 2019). Zhao et al. (2018) recognize performance degradation from non-i.i.d. data and propose global sharing of a small subset of data, which while effective, may compromise privacy. In settings of high statistical heterogeneity, fairness is also a natural question. AFL (Mohri et al., 2019) and $q$-FFL (Li et al., 2020) propose methods of focusing the optimization objective on the clients with the worst performance, though they do not change the network itself to model different data distributions.

### 3.2 PRESERVING PRIVACY

While much progress has been made in machine learning with public datasets (LeCun et al., 1998; Krizhevsky, 2009; Deng et al., 2009), in real-world settings, data are often sensitive, potentially for propriety (Sun et al., 2017), security (Liang et al., 2018), or privacy (Ribli et al., 2018) reasons. Protecting user data is one of the primary motivations for federated learning in the first place.

Approaches include releasing artificial data (Triastcyn & Faltings, 2020; Goetz & Tewari, 2020), homomorphic encryption (Hardy et al., 2017), or differential privacy (Dwork et al., 2006; Abadi et al., 2016; Melis et al., 2015). However, artificial data can still strongly resemble the original data, and sharing the model architecture and its parameters presents risks associated with whitebox access, leaving the data vulnerable to attacks such as membership inference (Shokri et al., 2017) or model inversion (Fredrikson et al., 2015; Wang et al., 2019; Zhu et al., 2019).

### 3.3 BAYESIAN NONPARAMETRIC FEDERATED LEARNING

Several previous works have applied Bayesian nonparameterics to federated learning, primarily as a means for parameter matching during aggregation. Instead of averaging the parameters weight-wise without considering the meaning of each parameter, past works have proposed using the Beta-Bernouilli Process (Thibaux & Jordan, 2007) for matching parameters, first with fully connected layers (Yurochkin et al., 2019), but later also extended by Wang et al. (2020) to convolutions and LSTMs (Hochreiter & Schmidhuber, 1997). In contrast, our method utilizes Bayesian nonparametrics for modeling rank-1 factors for multitask learning, instead of the aggregation stage.

## 4 EXPERIMENTS

### 4.1 EXPERIMENTAL SET-UP

Settings with higher statistical heterogeneity are more challenging for federated learning than when data are i.i.d. across clients, as well as more representative of the real-world, so we focus our experiments on the former. We consider two forms of statistical heterogeneity. The first is the simple non-i.i.d. construction introduced by McMahan et al. (2017), in which the data are sorted by class, sharded, and then randomly distributed to the $N$ clients such that each client only has data from $Z$ classes; many federated learning works consider the highly non-i.i.d. setting of $Z = 2$, which we also default to. While this setting can be challenging, it has the property that the classes present in every client's data is equally represented in the global data distribution. As a result, a single global model may perform reasonably uniformly across all clients. We thus refer to this as *unimodal* non-i.i.d.

However, this assumption of equal representation is generally not true in practice, as some characteristics or modes of the global distribution are inevitably less prevalent in the overall population than others. In the real world, this can correspond to age, gender, ethnicity, wealth, or a number of other demographic factors. To emulate this, we modify the above non-i.i.d. setting by first splitting the data and clients into two groups, with more clients in one group than the other. For MNIST (LeCun et al., 1998) for example, we partition the odd digits to 100 clients and the even digits to 20 clients. As before, each client still receives data from $Z = 2$ classes, with an equal number of data samples per client (the unallocated even digit samples are left unused); the difference is that there is now a $5 : 1$ ratio of odd to even digits in the total population, resulting in the clients with only even digits being in the minority of the global population. We call this setting *multimodal* non-i.i.d. Further details on the data allocation process and splits for FMNIST (Xiao et al., 2017) and CIFAR-10 (Krizhevsky, 2009) can be found in Appendix B.

In our experiments, the server selects a fraction $C = 0.1$ of clients during each communication round, with $T = 100$ total rounds for all methods. Each selected client trains their own model for $E = 5$

Table 1: Local Test Performance for $Z = 2$

| Dataset | Method | # of parameters ↓ | Unimodal ↑ | Multimodal ↑ |
|---------|--------|-------------------|------------|--------------|
| MNIST | FedAvg | 155,800 | 94.46±0.84 | 91.57±1.42 |
| | FedProx | 155,800 | 94.44±1.15 | 91.53±1.05 |
| | q-FFL | 155,800 | 91.46±1.07 | 88.42±1.24 |
| | WAFFLe | **120,200** | **96.23**±0.31 | **95.41**±0.36 |
| FMNIST | FedAvg | 28,880 | 83.96±0.91 | 83.43±2.27 |
| | FedProx | 28,800 | 84.19±0.99 | 83.59±2.30 |
| | q-FFL | 28,800 | 83.10±0.36 | 85.73±0.21 |
| | WAFFLe | **18,155** | **87.12**±0.89 | **86.09**±0.92 |
| CIFAR-10 | FedAvg | 61,770 | 52.54±0.14 | 45.46±1.69 |
| | FedProx | 61,770 | 52.36±0.11 | 44.95±1.17 |
| | q-FFL | 61,770 | 43.82±0.52 | 38.25±1.12 |
| | WAFFLe | **42,780** | **71.30**±0.92 | **66.35**±0.72 |

Table 2: Sub-population Local Test Performance Analysis

| Dataset | Method | Majority ↑ | Minority ↑ | Gap ↓ | Variance ↓ |
|---|---|---|---|---|---|
| MNIST | FedAvg | **96.63**±0.70 | 67.40±11.26 | 29.23±11.79 | 199±106 |
| | FedProx | 96.43±0.67 | 68.60±9.44 | 27.83±10.03 | 186±92 |
| | q-FFL | 94.93 ±0.31 | 54.20±7.37 | 40.73±7.55 | 355±117 |
| | WAFFLe | 95.93±0.16 | **93.87**±0.66 | **2.07**±0.77 | **26**±6 |
| FMNIST | FedAvg | 89.75±1.76 | 68.05±4.43 | 21.70±4.21 | 231±35 |
| | FedProx | **89.95**±1.73 | 67.50±4.50 | 22.45±4.38 | 233±42 |
| | q-FFL | 88.73±0.17 | 69.40±1.48 | 19.33±1.43 | 212±19 |
| | WAFFLe | 88.91±2.07 | **79.67**±1.52 | **9.25**±0.61 | **145**±27 |
| CIFAR-10 | FedAvg | 51.98±1.69 | 16.83±4.42 | 35.15±4.12 | 338±59 |
| | FedProx | 51.26±1.44 | 16.56±3.32 | 32.70±6.99 | 318±36 |
| | q-FFL | 42.00±0.29 | 18.14±3.05 | 23.87±3.00 | 220±17 |
| | WAFFLe | **68.37**±1.01 | **55.00**±6.00 | **13.37**±2.61 | **182**±27 |

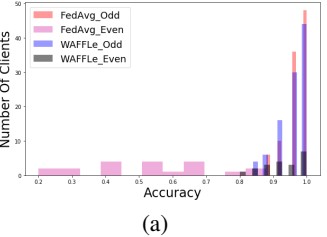 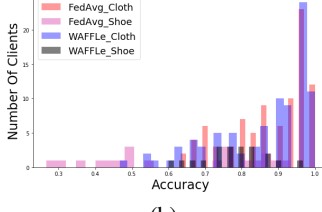 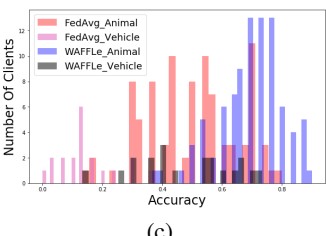

(a)                           (b)                           (c)

Figure 2: FedAvg and WAFFLe performance distribution across clients in the multimodal non-i.i.d. setting for (a) MNIST, (b) FMNIST and (c) CIFAR-10.

local epochs with mini-batch size $B = 10$, and the FedProx (Li et al., 2018) proximal parameter $\mu$ is set to 1.0. For q-FFL (Li et al., 2020), we searched $q \in \{0.001, 0.005, 0.01, 0.1, 1, 3, 5\}$ and found $q = 0.001$ as the best setting, matching the settings of Li et al. (2020) on more complex data. Model architectures, settings of $F$ and $\alpha$, and training schedules for each of the datasets are described in Appendix C. Ablation studies over the number of local epochs $E$, number of classes per client $Z$, and IBP parameters $\alpha$ and $F$ are provided in Appendix E, demonstrating robustness.

### 4.2 LOCAL TEST PERFORMANCE

We compare WAFFLe with FedAvg (McMahan et al., 2017), the fairness-oriented q-FFL (Li et al., 2020), and FedProx (Li et al., 2018), which augments FedAvg with a proximal term designed high statistical heterogeneity. We record local test performance averaged across all clients for both types of non-i.i.d. data allocation in Table 1, along with the total number of learnable parameters; plots of the training curves can be found in Appendix D. WAFFLe performs well despite strong statistical heterogeneity, as each client can learn a personalized model by selecting different factors from $\{W_a, W_b\}$; having a model specific to each data distribution results in higher local test performance than the baselines. This advantage is especially apparent when the data are distributed multimodal non-i.i.d., mainly because WAFFLe more effectively models underrepresented clients.

Interestingly, we find that WAFFLe outperforms the baselines particularly significantly for CIFAR-10, the most challenging of the tested datasets, with WAFFLe's local test performance outstripping the other methods by 18.8% and 20.9% for unimodal and multimodal settings, respectively. This demonstrates WAFFLe's ability to scale to complex tasks beyond MNIST, a common federated learning test bed. Notably, even though WAFFLe effectively learns a different model for each client, this does not lead to the computation or memory costs typically associated with independent models. WAFFLe's number of communication rounds is largely the same (Appendix D), and by sharing rank-1 factors, each weight factor can be represented compactly, resulting in a total number of parameters that is *fewer* than the single model used by the baselines, despite using the same architecture.

### 4.3 FAIRNESS

Average performance over all clients as in Table 1 is a commonly reported metric, but we argue that it does reveal the full story. We report subpopulation mean performance and overall population variance

in Table 2. We observe that FedAvg, which learns a single global model, focuses on minimizing mean error across the population, resulting in stronger performance for the clients in the majority. However, as a result, clients in the minority are severely compromised, as evidenced by the large difference ("Gap") between majority and minority values in Table 2; for example, FedAvg's performance for the "evens" group of clients is almost 30% lower than that of the "odds" group. This is gap is especially clear when visualizing the distributions of final local test performance for each client in the majority and minority groups (Figure 2). This underfitting can also be seen to exist throughout training from the "FedAvg_Minority" curve in Figure 6 of Appendix D, which lags far below the "FedAvg_Majority" in all three datasets. On the other hand, because of WAFFLe's shared weight factor dictionary design, different knowledge can be encoded in separate weight factors, which can be used by different parts of the population. As a result, despite certain classes being underrepresented (both in terms of clients, and total samples) in the training set, WAFFLe is able to successfully model them, with performances on par with the overall population. Notably, we achieve this without explicitly enforcing fairness through client sampling during training (Mohri et al., 2019; Li et al., 2020), which can be incorporated to further encourage uniform performance across clients.

## 4.4 Privacy Attacks

A primary objective of federated learning is to keep data safe. However, as mentioned in Section 2.3, the predominant federated learning strategy of each client sending their entire updated model's weights still leaves the client's data vulnerable. We demonstrate this with both membership inference and model inversion attacks.

Membership inference attacks (MIAs) (Shokri et al., 2017; Nasr et al., 2019) can be used to infer whether a given data query was used for model training, leveraging the tendency of machine learning to overfit or memorize training data. As such, a successful MIA can be used by an attacker to

Table 3: Membership Inference Attacks

| Methods | Accuracy | F1-score |
|---|---|---|
| FedAvg | 83.85± 1.62 | 83.72 ± 2.19 |
| WAFFLe | **56.20** ± 1.40 | **54.39** ± 1.85 |

surmise the content of a client's private data from the model. We compare a LeNet (LeCun et al., 1998) FedAvg (McMahan et al., 2017) model with an analogous WAFFLe model, training both on 1000 CIFAR-10 samples per client. We attack both with a MIA inspired by Shokri et al. (2017), using a small ensemble of 3 "shadow" models. As shown in Table 3, this simple attack achieves a high success rate at identifying a FedAvg client's training data, as intercepting the training update gives the full model. On the other hand, WAFFLe's training update only send partial model information, as the identity of the active factors is kept private. As a result, MIA success rate on WAFFLe is only moderately higher than random chance (50%). This means it is significantly harder to identify the private training data for WAFFLe, relative to FedAvg.

We also perform a model inversion attack (Fredrikson et al., 2015; Wang et al., 2019) on both FedAvg and WAFFLe. Unlike MIAs, which must start from a query data input, model inversion attacks seek to reconstruct the inputs used to train a model from the trained model itself; successful inversion attacks pose a significant risk from a data security perspective. We perform a model inversion attack on FedAvg and WAFFLe models trained on FMNIST, showing randomly selected results in Figure 3 recovered from an individual user. Importantly, reconstructions on FedAvg are significantly sharper, with the class identity far clearer than for WAFFLe, meaning FedAvg is more vulnerable to model inversion attacks.

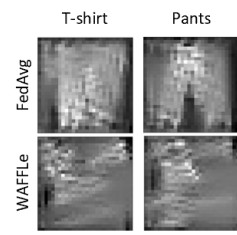

Figure 3: FMNIST model inversion attacks.

## 5 Conclusion

We have introduced WAFFLe, a Bayesian nonparametric framework for federated learning, employing shared rank-1 weight factors. This approach allows for learning individual models for each client's specific data distributions while still sharing the underlying learning problem in a parameter-efficient manner. Our experiments demonstrate that this model customizability makes WAFFLe successful at improving local test performance and, more importantly, significantly improves fairness in model performance when the data distribution among clients is multimodal. Furthermore, we are able to scale our results to CIFAR-10 and convolutional networks, where we observe the biggest improvements. We also show that by keeping the active factors selected by each model private on each device along with the data, WAFFLe's communication rounds only send partial model information, making it significantly harder to perform attacks on the private data.

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

## A  GENERALIZING WEIGHT FACTORIZATION TO CONVOLUTIONAL KERNELS

While introducing WAFFLe's formulation in Section 2.1, we assumed a multilayer perceptron (MLP) model, as illustrating our proposed shared dictionary with the 2D weight matrices composing fully connected layers is made especially clearer. While MLPs are sufficient for simple datasets such as MNIST, more challenging datasets require more complex architectures to achieve the most competitive results. For computer vision, for example, this often means convolutional layers, whose kernels are 4D. While 4D tensors can be similarly decomposed into rank-1 factors with tensor rank decomposition, such an approach would result in a large increase in the number of parameters in the weight factor dictionary due to the low spatial dimensions of the convolutional kernels (*e.g.*, $3 \times 3$) in most commonly used architectures. Instead, we reshape the 4D convolutional kernels into 2D matrices by combining the three input dimensions (number of input channels, kernel width, and kernel height) into a single input dimension. We then proceed with the formulation in (2). Similar approaches can be taken to generalize our formulation to other types of layers.

## B  DATA PARTITIONING

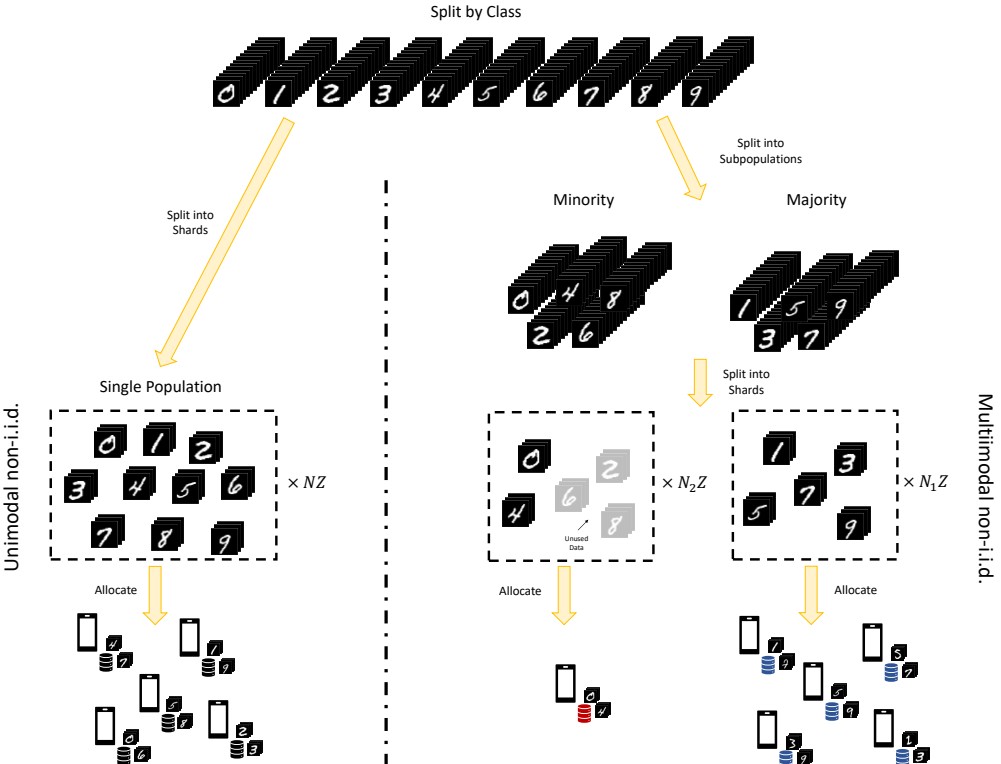

Figure 4: Example data allocation process to $N$ clients for MNIST and $Z = 2$ in the unimodal i.i.d. (left) and multimodal i.i.d. (right) settings. Notice that the primary difference is the grouping of the data into two subpopulations (here referred to as "Majority" and "Minority") before sharding and allocating $Z$ shards to each client.

Because statistical heterogeneity is an inherent property of federated learning paradigms, we focus our evaluation in this setting, testing WAFFLe in two different types of non-i.i.d. partitions of the data. A diagram showing the differences of the data allocation process for the two considered settings is shown for MNIST (LeCun et al., 1998) in Figure 4.

### B.1 UNIMODAL NON-I.I.D.

We first consider the non-i.i.d. setting introduced by McMahan et al. (2017). This is a widely used evaluation setting, commonly referred to as "non-i.i.d." or "heterogeneous" in other federated learning works, to distinguish it from completely i.i.d. data splits. We refer to this as *unimodal non-i.i.d.* to distinguish it from our second setting, which is also non-i.i.d. The primary purpose of such a partition is to investigate the behavior of federated average algorithms when each client has data from only a subset ($Z$) of classes.

This type of partition begins by sorting all data by class. Given $N$ client devices, the samples from each class are evenly divided into shards of data, each consisting of a single class, resulting in $NZ$ shards across all classes. These shards are then randomly distributed to the $N$ clients such that each receives $Z$ shards. The data in the $Z$ shards for each client is then shuffled together and split into a local training and test set. This ensures that the local test set for each client is representative of its own private data distribution.

### B.2 MULTIMODAL NON-I.I.D.

While the above partition does explore the non-i.i.d. nature of class distribution among clients, it does not adequately characterize the tendency for subpopulations to exist, with some being more prevalent than others. We propose a new non-i.i.d. setting to capture this, which we call *multimodal non-i.i.d.*, as each subpopulation group can be thought of as a mode of the overall distribution.

This partition begins similarly to unimodal non-i.i.d., with the data being sorted by class. Before sharding, however, classes are assigned to modes. The number of modes is arbitrary, but we choose two for simplicity, creating "majority" and "minority" subpopulations. In our experiments, the two modes are odd digits ($N_1 = 100$) versus even digits ($N_2 = 20$) for MNIST (LeCun et al., 1998), footwear and shirts ($N_2 = 20$) versus everything else ($N_1 = 90$) for FMNIST (Xiao et al., 2017), and animals ($N_1 = 90$) versus vehicles ($N_2 = 20$) for CIFAR-10 (Krizhevsky, 2009), where $N_1$ and $N_2$ are the number of clients in the majority and minority subpopulations, respectively. Once the classes have been separated by group, the process proceeds similarly to the unimodal i.i.d. partition process, with the data being divided into shards and then randomly allocated to clients within each subpopulation. We make the shards equal in size both within and across modes, so in instances where there are more data shards available than there are clients, we discard the unallocated data. Just as for unimodal non-i.i.d., local training and test sets are created for each client from its allocated data.

## C ADDITIONAL EXPERIMENTAL SET-UP DETAILS

**MNIST** For MNIST (LeCun et al., 1998) digit recognition, we use a multilayer perceptron with 1-hidden layer with 200 units using ReLU activations (Nair & Hinton, 2010). Based on this model, we constructed WAFFLe with $F = 120$ factors. The traditional 60K training examples are partitioned into local training and test sets as described in Section 4.1. Stochastic gradient descent (SGD) with learning rate $\eta = 0.04$ is employed for all methods.

**FMNIST** For FMNIST (Xiao et al., 2017) fashion recognition, we use a convolutional network consisting of two $5 \times 5$ convolution layers with 16 and 32 output channels respectively. Each convolution layer is followed by a $2 \times 2$ maxpooling operation with ReLU activations. A fully connected layer with a softmax is added for the output. Based on this model, we construct WAFFLe by only factorizing the convolution layers, with $F = 25$ factors. As with MNIST, the traditional 60K training examples are used to form the two local sets. SGD with learning rate $\eta = 0.02$ is used as the optimizer for all methods.

**CIFAR-10** For CIFAR-10 (Krizhevsky, 2009), we use we use a convolutional network consisting of two $3 \times 3$ convolution layers with 16 and 16 output channels respectively. Each convolution layer is followed by a $2 \times 2$ maxpooling operation with ReLU activations. These two convolutions are followed by two fully-connected layers with hidden size 80 and 60, with a softmax applied for the final output probabilities. To construct WAFFLe, we set the number of factors $F = 10$ for the two convolution layers, $F = 80$ for the first fully connected layer, and $F = 40$ for the second fully connected layer. The 50K training examples are used for constructing the local train and test sets. SGD with learning rate $\eta = 0.02$ is utilized for all methods.

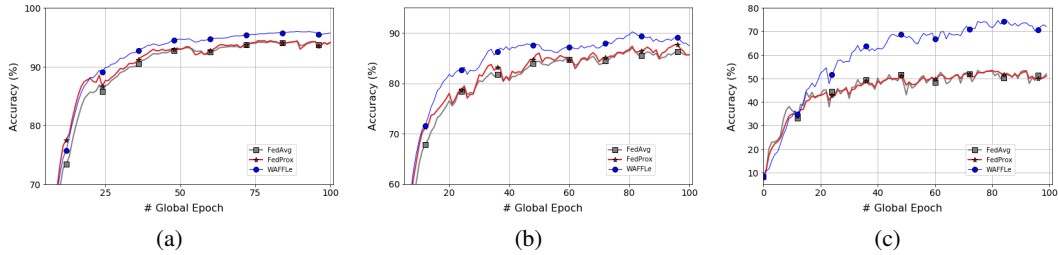

(a)           (b)           (c)

Figure 5: Local test performance for unimodal non-*i.i.d.* degree $Z = 2$. (a) MNIST; (b) FMNIST; (c) CIFAR-10.

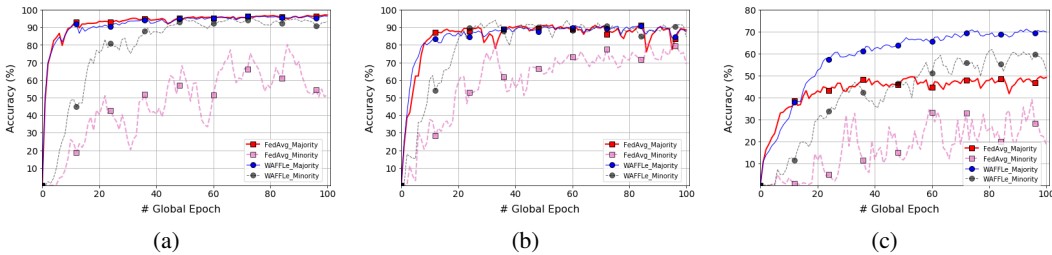

(a)           (b)           (c)

Figure 6: Local test performance for majority and minority subpopulations for multimodal non-*i.i.d.* degree $Z = 2$. (a) MNIST; (b) FMNIST; (c) CIFAR-10.

## D    TRAINING CURVES

We plot local test accuracy against the global epoch for FedAvg, FedProx and WAFFLe on MNIST, FMNIST, and CIFAR-10 averaged over all clients for unimodal non-*i.i.d.* data in Figure 5. A similar comparison is made between FedAvg and WAFFLe for multimodal non-*i.i.d.* data in Figure 6, with the majority and minority learning curves separately shown. For both cases, the clear gap between curves shows that WAFFLe achieves better performance throughout training. Notably, WAFFLe converges at a similar rate as FedAvg with respect to the global epoch number; this is important as the number of communication rounds is often considered one of the primary bottlenecks in federated learning.

In the multimodal non-*i.i.d.* case, the difference is especially stark for the minority subpopulation, which lags significantly behind the majority when modeled with FedAvg's one-size-fits-all approach. Interesting, in addition to having lower value, the FedAvg minority's training curve is not as smooth, with large dips and spikes, especially when compared with the majority subpopulation's curve. We hypothesize that this may be due to the smaller subpopulation being more vulnerable to being unrepresented during client sampling, which may lead to catastrophic forgetting (Shoham et al., 2019). We find this to be an interesting future direction of research. In comparison, the WAFFLe minority, with its separate set of customized weight factors, has a much smoother training trajectory.

## E    ABLATION STUDIES

**Statistical Heterogeneity** ($Z$)   WAFFLe is specifically designed for statistical heterogeneity, as each client can select different weight factors, effectively learning personalized models. WAFFLe was shown to excel when $Z = 2$, as this is a strongly non-i.i.d. setting: as each client only has samples from two classes. In Figure 7, we show how WAFFLe performs in unimodal settings with less statistical heterogeneity, for $Z = \{3, 4\}$. Although it takes longer to converge in these cases, WAFFLe still outperforms FedAvg by $7.20\%$ and $2.74\%$, respectively.

**Local epochs** ($E$)   Training client devices for more local epochs allows each server to collect a bigger update from each device, increasing local computation in exchange for fewer total communication rounds. This is often a desirable trade-off, as communication costs are commonly viewed as the primary bottleneck for federated learning. However, too many local epochs can lead to divergence

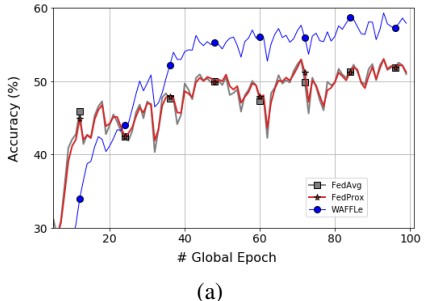 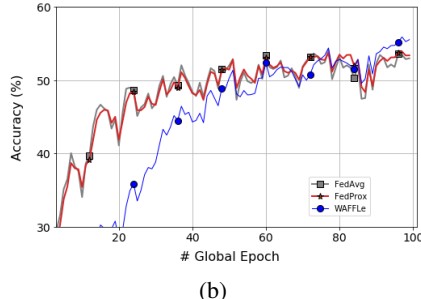

(a)                (b)

Figure 7: CIFAR-10 local test performance for statistical heterogeneity: (a) $Z = 3$; (b) $Z = 4$.

Table 4: Unimodal Local Test Accuracy vs Local Epochs

| Dataset | Method | E=10 | E=20 | E=30 |
|---------|--------|------|------|------|
| MNIST | FedAvg | 92.95 | 93.36 | 93.55 |
| | WAFFLe | 95.10 | 96.32 | 96.43 |
| FMNIST | FedAvg | 85.32 | 85.13 | 85.14 |
| | WAFFLe | 87.52 | 87.07 | 89.25 |
| CIFAR-10 | FedAvg | 47.40 | 47.60 | 55.39 |
| | WAFFLe | 64.18 | 71.92 | 74.50 |

Table 5: Multimodal Local Test Accuracy vs Local Epochs

| Dataset | Method | E=10 | E=20 | E=30 |
|---------|--------|------|------|------|
| MNIST | FedAvg | 88.70 | 89.27 | 89.03 |
| | WAFFLe | 95.37 | 94.87 | 95.07 |
| FMNIST | FedAvg | 86.21 | 86.58 | 86.47 |
| | WAFFLe | 87.03 | 89.15 | 91.33 |
| CIFAR-10 | FedAvg | 40.91 | 42.09 | 42.00 |
| | WAFFLe | 58.79 | 57.00 | 62.61 |

Table 6: Unimodal Local Test Accuracy vs $\alpha$ and $F$

| | F=80 | F=100 | F=150 |
|---|------|-------|-------|
| $\alpha/F = 0.4$ | 93.20 | 94.07 | 94.42 |
| $\alpha/F = 0.6$ | 95.08 | 94.48 | 95.56 |
| $\alpha/F = 0.8$ | 95.56 | 95.15 | 96.08 |
| $\alpha/F = 1.0$ | 96.33 | 95.63 | 96.45 |

Table 7: Multimodal Local Test Accuracy vs $\alpha$ and $F$

| | F=80 | F=100 | F=150 |
|---|------|-------|-------|
| $\alpha/F = 0.4$ | 91.83 | 92.70 | 93.23 |
| $\alpha/F = 0.6$ | 94.23 | 94.48 | 95.26 |
| $\alpha/F = 0.8$ | 94.76 | 95.15 | 95.70 |
| $\alpha/F = 1.0$ | 94.70 | 94.93 | 95.93 |

during the aggregation step. We study the influence of local epochs $E$ for unimodal non-i.i.d. in Table 4 and for multimodal non-i.i.d. in Table 5, using the same settings as in Section 4.1 except for reducing the global training epochs $T$ to 50 and the learning rate $\eta$ to 0.02 for all methods in multimodal non-i.i.d scenario. We observe that WAFFLe can handle increased number of local epochs, improving performance for all three datasets.

**Indian Buffet Process Sparsity ($\alpha$) and Number of Factors ($F$)** At the cost of more parameters, an increasing number factors $F$ and higher IBP parameter $\alpha$ gives client more expressivity for modeling its local distribution. We study the influence of $\alpha$ and $F$ for an MLP architecture on MNIST partitioned in both non-i.i.d. settings in Tables 6 and 7. As expected, the higher $\alpha$ and $F$ are, the better performance we observe, though in practice we prefer lower $\alpha$ and $F$ for efficiency. On the other hand, the overall difference in local test accuracy does not vary drastically, meaning that WAFFLe is fairly robust to both hyperparameters.

