# OpenReview forum: "WAFFLe: Weight Anonymized Factorization for Federated Learning"
_ICLR.cc/2021/Conference — Reject_

### Official Review · AnonReviewer2 · 2020-10-29
**Review of the Paper**

**Rating:** 5
**Confidence:** 5

**Review:**

-Summary-
The paper proposes a novel approach to federated learning which decomposes model parameters into two sub-modules with task-specific weight factors. The authors adopt the IBP process for sparse selections of the factors per client, which mitigates interferences across local clients. They validate the model with several baselines on two different non-iid settings.

-Pros-
- The proposed method largely outperforms baselines like FedAvg, FedProx, and p-FFL.
- The paper is easy to follow and intuition/direction is reasonable enough.
- Reasonable analysis of attacks and ablation studies (Appendix) is described.

-Cons-
- Lack of intuition about the architectural design of WAFFLe. There will be several similar designs like W_i^l \leftarrow \lambda_i^l * W_i^l (without decomposition) where \lambda_i^l is an attention vector, or r^l can leave as local without communication. Even weight factors can be regularized to l_0 norm.  I agree that the method looks reasonable but there need more insights into the necessity of model components.
- Only tiny networks are used. I recommend utilizing further modern complex CNN architectures. Applicability of modified network designs of WAFFLe on diverse modern CNNs is not guaranteed.
- Comparison with recent personalized FL/ Bayesian FL approaches is required. Since the method uses task-specific local parameters to mitigate inference from the naive aggregation of local parameters, the authors should compare with recent personalized FL/ Bayesian FL methods rather than old FL baselines.

---

> ### Author Response · Authors · 2020-11-23
> **Reviewer 2 Response**
>
> Thank you for the thoughtful feedback.
>
> C1. WAFFLe’s formulation is in part inspired by fairly standard rank-1 matrix factorizations and Bayesian factor analysis. As Reviewer 2 points out, our design choices are reasonable and intuitive. We agree that there may be different possible forms we could have picked, but that’s not necessarily the point of our paper. What’s more important is that we propose decomposing a federated learning model into shareable components in a principled manner, and we demonstrate the advantages of doing so. To address the specific alternatives suggested:
> - “$W_i^l \leftarrow \lambda_i^l * W_i^l$”: This can indeed provide personalization to each weight matrix, but it limits the number of “factors” to the dimensionality of $W_i^l$. Note that our formulation can grow the number of factors indefinitely, as needed as dictated by the data. Note the strong connections between our approach and the Indian Buffet Process, leading to a principled approach to factor assignment.
> - Keeping $r^l$ local: Yes, this is a potential design choice. However, we found empirically that initializing new clients from a global $r^l$ leads to better performance. We can add an ablation study for this if you think that’s necessary.
> - $L_0$ norm to weight factors: Presumably, this refers to imposing sparsity on the number of weight factors used per client. The $L_0$ norm isn’t differentiable, and REINFORCE type approaches have high variance. $L_0$ norm also doesn’t encourage weight factor reuse the way the Indian Buffet Process does, which we find to be an important consideration for federated learning.
>
> C2. We'd like to point out that the scale of our architectures are fairly standard for federated learning (e.g. see [1]). There are many recently published federated learning papers that only evaluate on synthetic datasets or MNIST, while we successfully scale to CIFAR-10. We've demonstrated with our experiments that our approach can be applied to convolutional kernels; how these kernels are arranged in an architecture is unlikely to have much impact on our model's feasibility.
>
> C3. To clarify, the Bayesian FL papers we cited [2,3] focus on using Bayesian nonparametrics to align locally learned features (which are normally permutationally invariant in deep networks) during federated aggregation. As such, they address a completely different problem from us (statistical heterogeneity), making comparisons not very insightful. We do agree with Reviewer 2’s suggestion to compare against a recent personalized federated learning method though. Thus, we implemented FedPer [4], showing results below on MNIST and CIFAR-10 in the multimodal non-i.i.d. setting. Note that we see superior performance from WAFFLe, particularly with the more challenging CIFAR-10.
>
> | Dataset | Majority  | Minority | Gap | Variance |
> |:-------------: |:---------------:|: -------------:|:----: | :----:|
> | MNIST     | 97.92(0.17)|   92.53(1.51)|  5.39(1.69)|  22(5)|
> | CIFAR-10     | 64.47(0.93 )|   36.02(2.60)|  28.41(1.72)|  237(38)|
>
> As a side comment, not all of our baselines are old. For evaluating fairness, we compare against q-FFL [5], which is from ICLR 2020.
>
> [1] LEAF: A Benchmark for Federated Settings. arXiv preprint arXiv:1812.01097
>
> [2] Bayesian Nonparametric Federated Learning of Neural Networks. arXiv preprint arXiv:1905.12022
>
> [3] Federated Learning with Matched Averaging. arXiv preprint arXiv:2002.06440
>
> [4] Federated learning with personalization layers. arXiv preprint arXiv:1912.00818
>
> [5] Fair Resource Allocation in Federated Learning. ICLR 2020

---

### Official Review · AnonReviewer3 · 2020-10-30

**Rating:** 4
**Confidence:** 4

**Review:**

This paper proposes WAFFLe for anonymized federated learning.  The idea seems interesting, but I have a few concerns.  In summary, the motivation/privacy claims are not clear, and the performance evaluation doesn't seem fair as the authors only considered single-model FL algorithms.

- Motivation (Why can't we just use SecAgg?): Secure aggregation guarantees that each local model parameter is protected.  See "Practical Secure Aggregation for Privacy-Preserving Machine Learning" by Bonawitz et al.  Thus, I am not sure about this problem's motivation.  Is there any reason one should employ this instead of secure aggregation?  (Note that the original secure aggregation protocol is computationally expensive, but the recent variations are not: For instance, see SecAgg+ [Bell et al., "Secure Single-Server Aggregation with (Poly)Logarithmic Overhead"] and TurboAGG [So et al., "Turbo-Aggregate: Breaking the Quadratic Aggregation Barrier in Secure Federated Learning"].)

- Missing privacy guarantees: While secure aggregation comes with a solid privacy guarantee, there is no theoretical guarantee that WAFFLe indeed can protect clients' data.  Especially with the IBP prior that induces sparsity, each client will update only a small subset of the weight factor dictionary.  This pattern may even reveal more information about the set of weight factors used by each client.  (Maybe the updated r also reveals this support information?)

- Non-adaptive attack: While the authors have presented some experimental results to claim improved privacy in Section 4.4, they only used the off-the-shelf attack algorithms.  The authors should have designed an adaptive attack algorithm, which could have performed much better.  The need for "adaptive evaluations" has been well described in [Tramer et al., "On Adaptive Attacks to Adversarial Example Defenses"], though in an adversarial example context.

- Multi-task/multi-center/personalization FL: The performance improvements in Table 1 and Table 2 seem mostly due to personalization, i.e., each model has its own model.  However, all the baseline algorithms assume a single model.  The authors should have compared the performance of WAFFLe with other algorithms that also maintain individual local models or multiple global models.  For instance, the authors may want to add MOCHA in [Smith et al., "Federated Multi-Task Learning"] (cited in the current work), and multi-center FL algorithms in [Sattler et al., "Clustered Federated Learning: Model-Agnostic Distributed Multi-Task Optimization under Privacy Constraints"] and its follow-up works.  The authors claimed that meta-learning-based approaches require sharing a small subset of data, but not all do.  For instance, see [Fallah et al., "Personalized Federated Learning: A Meta-Learning Approach"].

---

> ### Author Response · Authors · 2020-11-23
> **Reviewer 3 Response**
>
> Thank you for the insightful comments. Before we address them individually, we’d like to encourage taking a step back and viewing our work holistically. We’ve proposed a completely new approach for federated learning, which allows for a principled Bayesian nonparametric approach to per-client personalization while simultaneously (as a side effect of our construction) never transmitting the complete model, making it significantly harder to extract information on the client’s data. Reviewer 3 brings up some valid points, but we strongly disagree that they entirely discount our contributions, which we believe are still of value to the community.
>
> **Motivation:** Yes, SecAgg is also a way to secure federated learning; we can add discussion of SecAgg to our Related Works. As Reviewer 3 points out, the original forms of SecAgg are computationally expensive (quadratic for users, cubic for the server). While the recent works Reviewer 3 pointed out do reduce the expense to $O(n \log n)$, there are other weaknesses as well. For example, SecAgg protocols cannot handle adding new users without regenerating a new one. This can pose a significant problem for real-world federated settings, where users may come and go; by contrast, WAFFLe can gracefully expand its number of weight factors with the IBP as the number of clients grows, and clients leaving have no impact on the rest of the model. Additionally, while many SecAgg protocols are designed to be robust to curious or malicious attacks by clients, there are limits to how many such clients SecAgg can withstand, and increased security guarantees require more computation.
>
> That said, we still see SecAgg as an important component of security for federated learning. Security is often a multi-layered endeavor, and we see no reason why WAFFLe and SecAgg cannot be used in concert, nor why SecAgg makes WAFFLe’s contributions moot. We also want to emphasize that WAFFLe’s security aspects are not our only contribution. We initially designed WAFFLe to address statistical heterogeneity; the fact that WAFFLe can operate without transmitting complete model information is a positive side effect.
>
> **Missing privacy guarantees:** “each client will update only a small subset of the weight factor dictionary” <- This statement is not true. Because we use a soft relaxation of the Bernoulli distribution, the updates to unused factors are nonzero. While we do not give a privacy guarantee, we would like to point out that which weight factors are being used at a client is kept private; this information is not sent to the server by the client. Secure Aggregation is an orthogonal direction that provides a secure communication protocol and can very well be used with WAFFLe. These are not competing approaches.
>
> **Non-adaptive attack:** While it certainly may be the case that an attack custom designed for WAFFLe may do better, this doesn’t diminish the fact that WAFFLe is more resilient to off-the-shelf attacks, which we demonstrate FedAvg is vulnerable to. This is a win. Thoroughly attacking WAFFLe with a gauntlet of adaptive attacks is out of scope of this work, and best left to another paper.
>
> **Personalization:** Yes, Reviewer 3 is correct in pointing out that WAFFLe’s strong performance comes from being able to personalize client models. A comparison with other recently proposed personalized federated learning methods is thus a fair ask. However, we’d like to point out that the reason we do not compare with MOCHA despite citing it is "MOCHA does not apply to non-convex deep learning models" (from the MOCHA paper); we primarily consider deep learning models. Of the recently proposed personalized federated learning methods, we find FedPer [1] to be more related than the others suggested. We report results for FedPer on MNIST and CIFAR-10 in the multimodal non-i.i.d. setting below. Note that WAFFLe (Table 2) performs better.
>
> | Dataset | Majority  | Minority | Gap | Variance |
> |:-------------: |:---------------:|: -------------:|:----:  | :----:|
> | MNIST     | 97.92(0.17)|   92.53(1.51)|  5.39(1.69)|  22(5)|
> | CIFAR-10     | 64.47(0.93 )|   36.02(2.60)|  28.41(1.72)|  237(38)|
>
> [1] Federated learning with personalization layers. arXiv preprint arXiv:1912.00818

---

### Official Review · AnonReviewer1 · 2020-11-03
**Promising federated learning approach let down by unclear presentation and terminology**

**Rating:** 6
**Confidence:** 3

**Review:**

**Pros:**
+ Providing clients with additional flexibility to weight different portions of the globally learned weights allows for greater customization in federated learning
+ The client-specific hidden information leads to better test performance on non-iid data, along with better fairness and privacy properties for the trained models
+ Experiments are carried out across a range of datasets, with appropriate modifications made to simulate non-iid distribution among agents

**Cons:**
- The description of the IBP used to learn the local active factors in Section 2.2 is almost incomprehensible. The terms $v$ and $\pi$ are never defined, and it is unclear what the distribution $q$ corresponds to. While there may not be sufficient space in the main body to provide a detailed description, it should be included in the supplementary material without assuming detailed prior knowledge of Bayesian nonparametrics.
- Data privacy and security are often using interchangeably in the paper, when they correspond to very different concepts. This terminology needs to be clarified, as all the benefits of the proposed method are with regard to data privacy, and not security.

_Questions:_
1. Why is the number of learnable parameters smaller for CIFAR-10 than MNIST in Table 1?

---

> ### Author Response · Authors · 2020-11-23
> **Reviewer 1 Response**
>
> Thank you for the positive review!
>
> C1. We're sorry that the description of Section 2.2 wasn't clear. We used fairly well-established terminology from IBP: $v$ and $\pi$ are variational parameters used to construct the random variable $b$, and $q(r)$, $q(b)$ and $q(v)$ are the variational distributions. As suggested though, we will include a section in the Appendix that goes into more details for those without the appropriate background.
>
> C2. Thanks for the feedback. We will clarify our terminology in our next draft.
>
> Q1: The model used for MNIST was a fairly large 2 layer MLP; this isn't necessary to learn MNIST, but is a pretty commonly chosen architecture in federated learning. Note that the high dimensionality of the fully connected layers results in a large number of parameters. By contrast, our CIFAR model was a fairly compact CNN. Please see the model architecture descriptions in Section C of the Appendix for more details.

---

### Decision · Program_Chairs · 2021-01-07
**Final Decision**

**Decision:**

Reject

**Comment:**

This paper proposes an anonymization method for federated learning based on the Indian buffet process. The reviewers found the idea interesting, but raised the following main concerns (please see the reviews for more details):
* Motivation and terminology needs clarification
* Better comparison with secure aggregation methods
* Missing privacy guarantees
Overall the reviewers of this paper are borderline. I hope the authors will take the reviewers' feedback into account when revising the paper.